# Diquafosol Improves Corneal Wound Healing by Inducing NGF Expression in an Experimental Dry Eye Model

**DOI:** 10.3390/cells13151251

**Published:** 2024-07-25

**Authors:** Chieun Song, Hyemin Seong, Woong-Sun Yoo, Mee-Young Choi, Réka Dorottya Varga, Youngsub Eom, Seung Pil Yun, Seong-Jae Kim

**Affiliations:** 1Department of Ophthalmology, Institute of Health Sciences, College of Medicine, Gyeongsang National University, Gyeongsang National University Hospital, Jinju 52727, Republic of Korea; euni3494@naver.com (C.S.); seong_hm@daum.net (H.S.);; 2Department of Pharmacology and Convergence Medical Science, Institute of Health Sciences, College of Medicine, Gyeongsang National University, Jinju 52727, Republic of Korea; 3Department of Ophthalmology, Korea University College of Medicine, Seoul 02841, Republic of Korea; 4Department of Ophthalmology, Korea University Ansan Hospital, Ansan-si 15355, Republic of Korea

**Keywords:** dry eye disease, cyclosporine A, diquafosol tetrasodium, corneal epithelial cells, hyperosmolarity-induced, nerve growth factor

## Abstract

Dry eye disease (DED) is caused by inflammation and damage to the corneal surface due to tear film instability and hyperosmolarity. Various eye drops are used to treat this condition. Each eye drop has different properties and mechanisms of action, so the appropriate drug should be used according to clinical phenotypes. This study aims to compare the therapeutic mechanisms of cyclosporine A (CsA) and diquafosol tetrasodium (DQS). An experimental in vivo/in vitro model of DED using hyperosmolarity showed decreased cell viability, inhibited wound healing, and corneal damage compared to controls. Treatment with cyclosporine or diquafosol restored cell viability and wound healing and reduced corneal damage by hyperosmolarity. The expression of the inflammation-related genes *il-1β*, *il-1α*, and *il-6* was reduced by cyclosporine and diquafosol, and the expression of *Tnf-α*, *c1q*, and *il-17a* was reduced by cyclosporine. Increased apoptosis in the DED model was confirmed by increased Bax and decreased Bcl-2 and Bcl-xl expression, but treatment with cyclosporine or diquafosol resulted in decreased apoptosis. Diquafosol increased NGF expression and translocation into the extracellular space. DED has different damage patterns depending on the progression of the lesion. Thus, depending on the type of lesion, eye drops should be selected according to the therapeutic target, focusing on repairing cellular damage when cellular repair is needed or reducing inflammation when inflammation is high and cellular damage is severe.

## 1. Introduction

Dry eye disease (DED) is a multifactorial condition characterized by the loss of tear film homeostasis, leading to ocular discomfort and visual disturbance [1]. According to the Tear Film and Ocular Surface Society (TFOS) Dry Eye Workshop II (DEWS II) in 2017, DED involves tear film instability and hyperosmolarity, ocular surface inflammation, damage, and neurosensory abnormalities [2]. DED can be classified into two main types: aqueous tear-deficient dry eye, resulting from reduced tear production, and evaporative dry eye, due to excessive tear evaporation. Both types contribute to the inflammation and damage of the ocular surface, exacerbating the symptoms of DED [2,3]. A variety of eye drops are used to treat DED, including artificial tear solutions, sodium hyaluronate, autologous serum (containing growth factors), corticosteroids, tear composition stimulants (such as diquafosol tetrasodium), and immunomodulators (such as cyclosporine A). However, there are limitations, such as discomfort when using eye drops and poor efficacy [4].

Hyperosmolarity of the tear film is a significant mechanism in the pathophysiology of DED, leading to the activation of inflammatory pathways and damage to the corneal epithelium [5]. Inflammatory cytokines, such as interleukins (il-1β, il-1α, and il-6) and tumor necrosis factor-alpha (Tnf-α), are upregulated in response to hyperosmolar stress, contributing to cell death and impaired wound healing [5,6]. The imbalance between pro-inflammatory and anti-inflammatory signals exacerbates the chronic inflammation observed in DED [6,7].

Various pharmacological treatments have been employed to manage DED, including artificial tears, immunomodulators, and tear secretion promoters [8]. Cyclosporine A (CsA) is a potent immunomodulator inhibiting T-cell activation and downregulating inflammatory cytokines, thereby reducing ocular surface inflammation [8,9]. CsA has been extensively used to treat DED by improving tear film stability and reducing inflammation [9,10]. On the other hand, diquafosol tetrasodium (DQS) is a dinucleotide derivative that acts as an agonist of the purinergic P2Y2 receptor [11]. DQS stimulates the secretion of mucins and water from conjunctival epithelial cells and accessory lacrimal glands, enhancing tear film stability and protecting the ocular surface from desiccation [12,13].

Recent studies have highlighted the potential of CsA and DQS in treating DED [14], but the molecular mechanisms by which these drugs alleviate cellular damage and promote corneal wound healing are not fully understood. Notably, nerve growth factor (NGF), a neurotrophin involved in neuronal survival and differentiation [15], has been implicated in corneal epithelial cell migration, proliferation, and wound healing [15,16]. NGF binds to its high-affinity receptor TrkA, initiating signaling cascades that promote cell survival and inhibit apoptosis [17].

This study aims to elucidate the mechanisms by which CsA and DQS influence corneal wound healing and the development of ocular surface damage in an experimental dry eye model induced by hyperosmolarity. We hypothesize that CsA and DQS modulate NGF expression and signaling, thereby enhancing corneal epithelial cell survival, reducing apoptosis, and promoting wound healing. By comparing the effects of CsA and DQS on inflammatory cytokine expression, apoptosis markers, and NGF signaling, we seek to provide insights into the differential therapeutic mechanisms of these drugs and their potential for improving DED treatment outcomes.

## 2. Materials and Methods

### 2.1. Cell Culture and Treatment

Human corneal epithelial cells (HCECs), an HCEC line immortalized by the viral transformation of SV-40, were purchased from the American Type Culture Collection (Manassas, VA, USA). These cells were maintained in a keratinocyte serum-free medium (Keratinocyte-SFM; Gibco, Thermo Fisher Scientific, Inc., Waltham, MA, USA) containing 1× human corneal growth supplement (HCGS; Gibco), 100 units/mL of penicillin (Gibco) or 100 µg/mg of streptomycin (Gibco) in 5% CO_2_ at 37 °C. In the entire experiment, when only media were used, they were described as control media (CTL), and hyperosmolar stress was applied by adding 350, 400, 450, 500, and 550 mOsm/L of NaCl to this baseline media for 24 h. For hyperosmotic stress at different time points, sterile sodium chloride (1 M) was added at 500 mOsm/L to the culture media for 4, 8, 16, and 24 h.

### 2.2. Treatment of HCECs with DQS and CsA

HCECs were seeded in 6-well plates at a density of 2 × 10^5^ cells/mL with the addition of 10, 50, 100, 200, and 400 μM of DQS (Sigma-Aldrich, St. Louis, MO, USA) or 0.1, 0.5, 1, 5 and 10 μM CsA (Sigma-Aldrich) for 8 h, depending on the experiments. For the effect of DQS and CsA in HS, HCECs were seeded in 24-well plates at a density of 1 × 10^5^ cells per well and incubated for 24 h. The next day, the cells were cultured in a medium (CTL) containing 500 mOsm/L of NaCl (HS; hyperosmotic stress) and 100 μM of DQS (HS + DQS) or 1 μM of CsA (HS + CsA) for 8 h.

### 2.3. Cell Viability Assay

Cell viability was measured using the MTT assay. Briefly, HCECs were inoculated into 24-well plates at 5 × 10^4^ cells/mL per well and incubated for 24 h. The media were aspirated and replaced with media containing different treatments with NaCl (350, 400, 450, 500 (HS), and 550 mOsm/L), DQS (10, 50, 100, 200, and 400 μM), CsA (0.1, 0.5, 1, 5, and 10 μM), HS + DQS (500 mOsm/L of NaCl with 100 μM of DQS) and HS + CsA (500 mOsm/L of NaCl with 1 μM of CsA) for 24 h, followed by 4 h of incubation with 3-(4,5-dimethylthiazol-2-yl)-2,5-diphenyltetrazolium bromide (MTT; Sigma-Aldrich). The MTT-transformed crystals were dissolved in DMSO, and the absorbance at 570 nm was measured using an ELISA reader (Molecular Devices, Sunnyvale, CA, USA).

### 2.4. Quantitative Real-Time PCR

Total RNA was extracted from HCECs and mouse corneal tissue at the indicated time points, and first-strand cDNA was synthesized using random hexamer primers provided in the first-strand cDNA synthesis kit (Applied Biosystems, Framingham, MA, USA) according to the manufacturer’s instructions. Total RNA (1 µg) was used for cDNA synthesis on an iCycler thermocycler (Bio-Rad, Hercules, CA, USA). qPCR was performed using an iQ SYBR Green Supermix kit (Bio-Rad) with a CFX Connect Real-Time PCR Detection System (Bio-Rad). PCR primers were synthesized based on reported cDNA sequences in the NCBI data bank. Sequences of the primers used for PCR were demonstrated following the set of primers (sense and antisense, respectively) in Table 1.

### 2.5. Wound Healing Assay

For the wound healing assay, cell migration was detected. HCECs were seeded in a two-well culture-insert cassette (ibidi culture-insert 2-well; ibidi GmbH, Martinsried, Germany) at a density of 2.5 × 10^4^ cells per well. After the cells were allowed to adhere for 24 h, the culture-insert cassette was removed and washed with PBS to remove nonadherent cells. Fresh CTL, HS, HS + DQS, and HS + CsA media were then added, and the plate was photographed at 0 and 8 h to capture the two different fields at each time point on each plate. The number of cells migrating into the wound space was manually counted in three fields per well under a light microscope. Quantification of the areas was performed using ImageJ software (Ver. 1.51f51, NIH, Bethesda, MD, USA).

### 2.6. Western Blot Analysis

HCECs were lysed in RIPA lysis buffer with a Halt protease and phosphatase inhibitor cocktail (Thermo Scientific, Inc., Waltham, MA, USA), sonicated, and centrifuged at 12,000× *g* for 10 min at 4 °C to remove insoluble debris. Protein concentrations of the cell lysates were determined using a BCA Protein Assay Kit (Pierce Biotechnology, Rockford, IL, USA). Whole-cell lysates were separated by SDS–PAGE in a 10% polyacrylamide gel and transferred to a nitrocellulose membrane (Millipore, Bedford, MA, USA). After blocking with 5% nonfat dry milk, each blot was incubated with primary antibodies against B-cell lymphoma 2 (Bcl-2; Santa Cruz Biotechnology, Santa Cruz, CA, USA), B-cell lymphoma-extra large (Bcl-xL; Santa Cruz Biotechnology), Bcl-2-associated X (Bax; Santa Cruz Biotechnology), glutathione peroxidase (GPx; Santa Cruz Biotechnology), NGF (Abcam, Cambridge, UK), and β-actin (Sigma-Aldrich), and then with horseradish peroxidase-conjugated anti-rabbit immunoglobulin (Ig) G or anti-mouse IgG (Cell Signaling Technology, Danvers, MA, USA). Antibody binding was detected using SuperSignal Chemiluminescent Substrate (Thermo Fisher Scientific). Images were acquired using the ChemiDoc Touch Imaging System (Bio-Rad). Densitometry was performed using ImageJ software (NIH).

### 2.7. ELISA

Human NGF levels were measured in HCECs using an NGF ELISA kit (Cusabio Technology, Wuhan, China). Briefly, the NGF ELISA kit was designed for the sensitive and specific detection of NGF in the antibody sandwich format. HCECs were inoculated at 5 × 10^4^ cells/mL per well in 24-well plates and incubated for 24 h. The media were aspirated and replaced with media containing different treatments with HS, HS + DQS, and HS + CsA for 8 h. The supernatants of HCECs after differentiation were collected, and 100 μL supernatants were transferred into flat-bottom 96-well plates coated with an anti-NGF polyclonal antibody (pAb), which binds soluble NGF. The captured NGF was bound by a biotin antibody. After washing, the amount of specifically bound biotin antibody was detected using a species-specific antibody conjugated to horseradish peroxidase (HRP) as a tertiary reactant. The unbound conjugate was removed by washing, and following incubation with a chromogenic substrate, the color change was measured using a spectrophotometer using an ELISA reader (Molecular Devices). The amount of NGF in the test solutions is proportional to the color generated in the oxidation-reduction reactions. NGF is expressed as pg/mL.

### 2.8. Animal Model and Treatment

The animal experiments were carried out following the National Institutes of Health Guidelines on the Use of Laboratory Animals and the protocol approved by the Gyeongsang National University Institutional Animal Care and Use Committee for Animal Research (GNU-220602-M0061). The experimental DED test was performed on 6- to 8-week-old C57BL/6 mice, as described by Li et al. [14]. Three groups of mice were instilled with 1 μL of hyperosmolar saline (HS; 500 mOsm/L) in both eyes six times a day for 10 days. Two of these groups received 1 μL of 3% dicaposol tetrasodium (DIQUAS^®^, Santen Pharmaceuticals, Osaka, Japan) or 0.05% cyclosporine (CyporinN, Taejun Pharmaceuticals, Seoul, Republic of Korea) 30 min before HS instillation in both eyes for 10 days starting on Day 10. Age-matched phosphate-buffered saline (PBS)-treated mice were used as controls (CTL). Four to six mice per group were evaluated in each experiment. All mice were euthanized by a veterinarian. Experiments were performed at least three times to verify the reproducibility of the results.

### 2.9. Corneal Fluorescein Staining

The cornea was examined using a slit-lamp biomicroscope by instillation of 1 µL of 5% fluorescein for 3 min with a micropipette into the inferior conjunctival sac of both eyes. Punctate corneal staining was quantified using ImageJ software (NIH).

### 2.10. Immunofluorescence Staining

For immunofluorescence staining, after the experimental dry eye test, corneas were dissected and then fixed in 4% paraformaldehyde (PFA) overnight at 4 °C. After washes with fresh 1× PBS, corneas were soaked in 30% sucrose for 24 h, embedded in optimum cutting temperature (OCT) compound (Tissue-Tek^®^, Sakura Fine Technical, Tokyo, Japan), and stored at 80 °C. Ten-micrometer cryostat sections were placed on silane-coated slides (5116-20F, Muto Pure Chemical, Tokyo, Japan) and kept at 37 °C for 4 h to air dry. Samples on slides were rinsed in PBS and permeabilized in PBS with 0.1% Triton-X (PBST) containing 5% donkey serum for 1 h at RT. The slides were incubated with primary antibodies in dilution buffer (0.1% PBST) overnight at 4 °C and subsequently incubated with secondary antibodies in dilution buffer for 1 h at RT. The slides were rinsed three times with 0.1% PBST after each step. The slides were mounted in mounting medium with 4′,6-diamidino-2-phenylindole (DAPI; Vectashield^®^, Vector Laboratories, Inc., Burlingame, CA, USA) and examined with a confocal laser scanning microscope (LSM710, Carl Zeiss Meditec AG, Jena, Germany). Rabbit anti-NGF (1:500; Abcam) was used as the primary antibody. Secondary antibodies (1:500; Thermo Fisher Scientific) were matched to the host species of the primary antibodies.

### 2.11. Statistical Analysis

All data were analyzed using the GraphPad Prism 6 software (San Diego, CA, USA). Data are presented as the mean ± standard error of the mean (S.E.M.) of quantifiable results from at least three independent experiments. Statistical analysis was determined by unpaired two-tailed Student’s *t* test or one-way analysis of variance (ANOVA) followed by Tukey’s multiple comparison test. Assessments with *p* < 0.05 were considered statistically significant.

## 3. Results

### 3.1. Hyperosmolarity-Induced Cellular Damage in Human Corneal Epithelial Cells

The cultured corneal epithelial cells (HCECs) were treated with 350 to 550 mOsm/L of NaCl and incubated for 24 h to observe morphological changes, which were examined by phase-contrast microscopy (Figure 1A). At a concentration of 500 mOsm/L, the shape of the HCECs became rounded, and the number of cells detached from the culture dish increased.

Cell viability was measured using MTT assays, and a 50% reduction in cell viability was observed at 500 mOsm (Figure 1B). When cell viability was measured at various times at a concentration of 500 mOsm/L, there was a slight decrease at 8 h, but a 50% decrease at 16 h was significant (Figure 1C). Based on this result, we performed further experiments with an 8 h treatment of 500 mOsm, which has a hyperosmolar effect on cells but does not cause cell death.

### 3.2. Effects of Diquafosol Tetrasodium and Cyclosporine in Hyperosmolar Treatment

The cell viability of HCECs treated with DQS and CsA at various concentrations showed almost CTL-like cell viability up to 100 µM in DQS and 1 µM in CsA (Figure 2A,B). Treatment with 100 µM of DQS and 1 µM of CsA with 500 mOsm hyperosmolarity (HS) did not reduce cell viability as significantly as HS treatment. These results indicate that DQS and CsA can inhibit HS-induced cell damage (Figure 2C).

### 3.3. Changes in Inflammatory Cytokine Expression by DQS and CsA with HS Treatment

We measured changes in inflammatory cytokines using qPCR in HCECs after 8 h of exposure to HS treatment or DQS or CsA in combination with HS treatment. The expression of *il-1β*, *il-1α*, and *il-6* was similar to that of the CTL in the HS+DQS- and HS + CsA-treated cells compared to HS-treated cells (Figure 3A–C). In particular, the expression of *Tnf-α*, *c1q*, and *il-17a* was slightly reduced in the HS+DQS-treated cells compared to that in the cells treated with HS alone; however, in the HS + CsA-treated cells, the expression level was similar to that in the CTL group (Figure 3D–F). The expression level of *nf-kb*, which is known to regulate the expression of inflammatory cytokines, was approximately three times higher in the HS-treated cells than in the CTL cells. However, when DQS or CsA was cotreated with HS, the expression level was similar to that in the CTL cells (Figure 3G).

### 3.4. Effect of DQS and CsA on Wound Healing in HS-Treated Cells

We measured wound healing after 8 h of exposure to HS, HS + DQS, and HS + CsA (Figure 4). Exposure to HS resulted in delayed wound healing, but treatment with DQS or CsA accelerated wound healing. In particular, treatment with diquafosol resulted in wound healing at a similar level to CTL, in contrast to cyclosporine.

### 3.5. Effect of DQS and CsA on Apoptosis after Hyperosmotic Stress Exposure

The expression of the apoptosis-related proteins Bax and Bcl-XL in HCECs exposed to hyperosmolarity was examined by Western blotting. The protein level of Bax increased upon exposure to HS, but after treatment with DQS or CsA, the expression of Bax was similar to that of the CTL group (Figure 5A,D). The expression of Bcl-2 and Bcl-xl increased more after treatment with DQS or CsA than after exposure to only HS (Figure 5A–C). The Bax/Bcl-2 ratio was also higher with HS exposure than with CTL, HS+DQS, or HS+CsA (Figure 5E).

To investigate the effect of DQS and CsA on hyperosmolarity-induced oxidative stress, we confirmed the expression of GPx, an antioxidant enzyme. We observed that the protein level of GPx decreased in the HS-exposed cells. After treatment with both DQS and CsA, the level of this protein expression was similar to the CTL (Figure 5F).

### 3.6. Regulation of NGF Expression by DQS and CsA In Vivo and In Vitro

Examination of the effects of DQS or CsA on NGF signaling during hyperosmotic exposure showed that HS increases NGF protein levels and the p-Trk-A/t-Trk-A expression ratio. However, after exposure to HS+DQS or HS + CsA, the p-Trk-A/t-Trk-A expression ratio and NGF protein levels were similar to or lower than those in the CTL group (Figure 6A–C). The RNA level of NGF was found to be highest in HCECs exposed to HS+DQS and lowest in those exposed to HS and HS+CsA (Figure 6D). The NGF level was measured by ELISAs and was approximately 2-fold higher in the HS + DQS group than in the HS group; in the HS + CsA group, the amount was similar to that of the CTL group (Figure 6E).

### 3.7. Effect of DQS and CsA on Corneal Staining in Mice

To investigate the effect of DQS and CsA on corneal wound healing in HS-exposed DED mice, we obtained slit lamp images in 4 groups (Figure 7A). The corneal fluorescein staining scores (NEI) significantly increased compared to CTL in the HS group, and treatment with DQS and CsA resulted in decreased corneal staining (Figure 7B).

### 3.8. Effects of DQS and CsA on the Inflammatory Response and ngf Expression in an Experimental Dry Eye Model

In an experimental dry eye model, we found that treatment with DQS and CsA altered the expression levels of inflammatory cytokine genes. After exposure to HS, the expression levels of *il-1β*, *Tnf-α*, *il-6*, *il-17a*, and *il-10* were higher than those of the CTL group (Figure 8A–H). When DQS and CsA were instilled after HS, the expression levels were similar to those of the CTL group (Figure 8A–E). The expression of *ngf* increased only in the mice treated with HS + DQS (Figure 8I). Immunofluorescence staining in the cornea showed increased *ngf* expression in the corneal epithelium of the HS-treated mice compared to CTL. In particular, the HS + DQS-treated mice showed more *ngf* expression than the HS-only or HS + CsA-treated mice (Figure 9).

## 4. Discussion

Dry eye disease (DED) is a complex, multifactorial condition characterized by a loss of tear film homeostasis, ocular surface inflammation, and damage [2]. The condition can be classified into two primary types: aqueous tear-deficient dry eye, resulting from decreased tear production, and evaporative dry eye, caused by excessive tear evaporation. Both types lead to increased tear osmolarity, triggering inflammation and subsequent damage to the ocular surface [2,3]. Given the heterogeneous nature of DED, it is crucial to tailor treatments based on the specific clinical phenotypes of the disease [5]. Our study explored the therapeutic mechanisms of CsA and DQS in an experimental dry eye model induced by hyperosmolarity. Hyperosmolar stress is a key driver in DED pathogenesis, leading to an inflammatory cascade and corneal surface damage [5]. CsA is a potent immunomodulator that inhibits T-cell activation and downregulates several inflammatory cytokines, thereby reducing ocular surface inflammation. Its efficacy in treating inflammatory ocular surface diseases was first reported by Laibovitz et al. [7]. Topical CsA has been shown to reduce cell-mediated inflammatory responses, improving tear film stability and reducing ocular surface inflammation [8,9]. DQS, on the other hand, is a dinucleotide derivative that functions as an agonist of the purinergic P2Y2 receptor [11], stimulating mucin and water secretion from conjunctival epithelial cells and accessory lacrimal glands. This action helps to stabilize the tear film and protect the ocular surface from desiccation [12,13]. Our findings highlight that CsA and DQS exhibit distinct effects on DED, suggesting that their clinical applications should be differentiated based on the underlying pathophysiology and phenotype of the disease.

Our findings demonstrate that both CsA and DQS mitigate hyperosmolarity-induced cellular damage by modulating inflammatory responses and promoting corneal epithelial cell survival. As previously published papers have stated, both CsA and DQS decrease the cell viability of HCECs at higher concentrations [18,19]. We conducted our experiments using doses that, according to previous studies, do not significantly affect cell viability. Treatment with either CsA or DQS restored cell viability and reduced inflammatory cytokine expression in human corneal epithelial cells (HCECs) exposed to hyperosmotic stress. However, the two drugs exhibited differential effects on specific cytokines. For instance, CsA reduced the expression of *Tnf-α*, *c1q*, and *il-17a* more effectively, indicating its superior efficacy in suppressing the inflammatory response compared to DQS. CsA is currently the most widely used immunosuppressive drug due to its ability to inhibit cell-mediated inflammatory responses induced by interleukin-2 (il-2) and interferon-gamma (IFN-γ) and its mechanism of inhibiting conjunctival cell death [16]. The anti-inflammatory function of CsA has been shown to prevent inflammation-induced apoptosis, thereby protecting conjunctival epithelial cells and reducing the enhanced inflammatory response in conjunctival epithelial cells exposed to hyperosmolarity [17]. Consistent with previous studies showing that DQS has the potential to improve immune-related disease in corneal disease by reducing the expression of Tnf-α and il-6 [20], exposure of HCECs to HS+DQS reduced the expression of several immune response cytokines (*Tnf-α*, *il-6*, *il-1β*, and *il-1α*) more than HS alone. Apoptosis was another critical aspect examined in our study. Hyperosmotic stress increased the expression of pro-apoptotic Bax and decreased the expression of anti-apoptotic Bcl-2 and Bcl-xL, leading to heightened apoptosis in HCECs. Both CsA and DQS reduced Bax expression and increased Bcl-2 and Bcl-xL expression, thus lowering the Bax/Bcl-2 ratio and preventing apoptosis. This protective effect against apoptosis underscores the therapeutic potential of CsA and DQS in preserving corneal epithelial cell integrity under hyperosmotic conditions.

A novel finding in our study is the role of nerve growth factor (NGF) in corneal wound healing and its regulation by CsA and DQS. NGF is a neurotrophin that plays a key role in neuronal survival, differentiation, and prevention of apoptosis [15,16]. NGF promotes the activation of signaling pathways, including ERK1/2 and Akt, which are crucial for cell survival and proliferation [17]. Our results show that DQS significantly enhances NGF expression and its release into the extracellular space, thereby promoting corneal epithelial cell migration and wound healing. This effect of DQS on NGF expression suggests that DQS may be particularly beneficial in promoting corneal repair processes through NGF-mediated mechanisms. In vivo experiments in an experimental dry eye mouse model further supported our in vitro findings. Treatment with DQS and CsA improved corneal damage scores, reduced inflammatory cytokine levels, and modulated NGF expression. Notably, DQS-treated mice exhibited higher levels of NGF expression than CsA-treated mice, highlighting the potential of DQS in enhancing corneal wound healing through NGF-mediated pathways.

NGF and its receptor are expressed in corneal epithelial cells, and NGF expression was reported to increase when corneal epithelial cells are damaged, suggesting that NGF plays an important role in corneal epithelial cell migration and proliferation [21,22]. In the wound healing assay, the wound healing of the HCECs exposed to HS was weak, whereas wound healing was restored by exposure to HS + DQS or HS + CsA. In particular, the HCECs exposed to HS + DQS recovered almost as well as the CTL cells. In the HS-treated HCECs, NGF protein expression levels and Trk-A phosphorylation were increased compared to those of the CTL group. However, after exposure to HS+DQS or HS + CsA, CTL-like protein expression levels were confirmed. In contrast to the protein level, the RNA level was increased compared to that of the CTL group upon HS or HS + CsA exposure, with a greater increase upon HS+DQS exposure. When the amount of extracellular NGF was measured by ELISAs, the amount released from the HCECs exposed to HS + DQS was more than 2-fold higher than that released from the HCECs exposed to HS or HS + CsA. After exposure to HS and HS + DQS, the level of intracellular *ngf* RNA expression was highest, but the level of NGF protein expression was similar to that of the CTL group. Further research is needed to understand the mechanism by which DQS facilitates the transport of increased NGF expression out of the HCECs exposed to HS. Since NGF plays a role in regulating cell migration and signaling to prevent apoptosis [15,16], it is suggested that the increased expression of NGF by DQS was released to the outside of the cell, enhancing wound healing and preventing apoptosis. P2Y2 receptors are involved in neuronal development and regeneration, and NGF induces the activation of TrkA, which activates a signaling cascade leading to the activation of ERK1/2 [23]. In addition, during wound healing in DQS-treated corneal epithelial cells, cell proliferation and migration occur through the activation of ERKs via DQS-mediated increases in Ca^2+^ [24]. Although some studies have shown that the activity of the P2Y2/TrkA signaling pathway is increased in the presence of NGF, there are no studies showing that P2Y2 regulates NGF production. Our results show that DQS, a P2Y2 receptor agonist, increases the expression of NGF, thereby increasing the amount of NGF present in the extracellular space. Despite these results, it is still unknown how P2Y2 or P2Y2 receptor agonists activate the NGF transcriptional mechanism, which ultimately increases the amount of NGF inside and outside the cells, and further research is required.

However, our study has the following limitations. First, we used a corneal epithelial cell line. Second, we used a hyperosmolar solution as an in vivo dry eye model. Third, we did not investigate the mechanism by which DQS facilitates the secretion of increased NGF expression from HCECs. Future research will utilize primary cultured cells and well-known mouse models such as desiccating stress or scopolamine injection to study the mechanisms involved.

## 5. Conclusions

CsA and DQS have distinct roles in the management of DED, with CsA being more effective at reducing inflammation and DQS at promoting epithelial healing and tear secretion. The choice of therapy should be guided by the specific clinical phenotype of DED, ensuring that the treatment addresses the underlying pathophysiology of the disease. Future studies should further explore combination therapies and personalized treatment approaches to optimize outcomes for DED patients.

## Figures and Tables

**Figure 1 cells-13-01251-f001:**
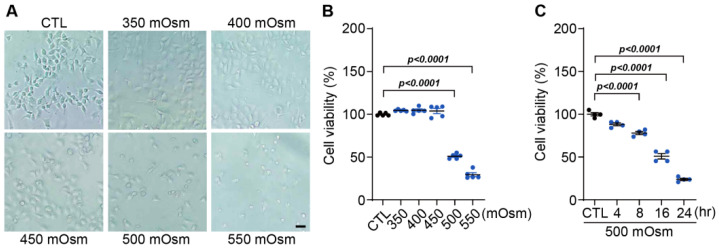
Effect of hyperosmotic stress on the viability of human corneal epithelial cells (HCECs) (**A**) HCECs were cultured in a medium with various osmolarities (control media (CTL), 350, 400, 450, 500, and 550 mOsm/L of NaCl). Representative phase contrast images were obtained (scale bar = 25 μm). (**B**) The effect of hyperosmotic stress on the activity of mitochondrial dehydrogenases in HCECs was measured by the MTT assay. (**C**) Cell viability was determined by MTT assays in HCECs cultured with 500 mOsm/L of NaCl for various time points (0 (CTL), 4, 8, 16, and 24 h). The viability of control cells (control media, CTL) was set to 100%. Viability is shown as a percentage of that of the control cells. Bars represent the mean ± S.E.M. (*n* = 5).

**Figure 2 cells-13-01251-f002:**
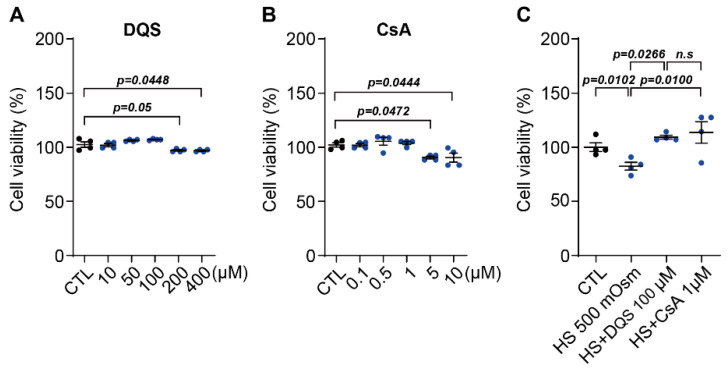
Effect of DQS and CsA on hyperosmotic stress-induced cell death. (**A**,**B**) HCECs were treated with various concentrations of DQS ((**A**); 0 (CTL), 10, 50, 100, 200, and 400 μM/L of DQS) and CsA ((**B**); 0 (CTL), 0.1, 0.5, 1, 5, and 10 μM/L of CsA) in medium for 8 h. Cell viability was measured by MTT assays (*n* = 4). (**C**) HCECs were cotreated with DQS (HS + DQS 100 μM) and CsA (HS + CsA 1 μM) in media with an osmolarity of 500 mOsm/L NaCl (HS) for 8 h. CTL cell viability was set to 100%. Viability is shown as a percentage of that of the control cells. Bars represent the mean ± S.E.M. (*n* = 4); *n.s*: nonsignificant.

**Figure 3 cells-13-01251-f003:**
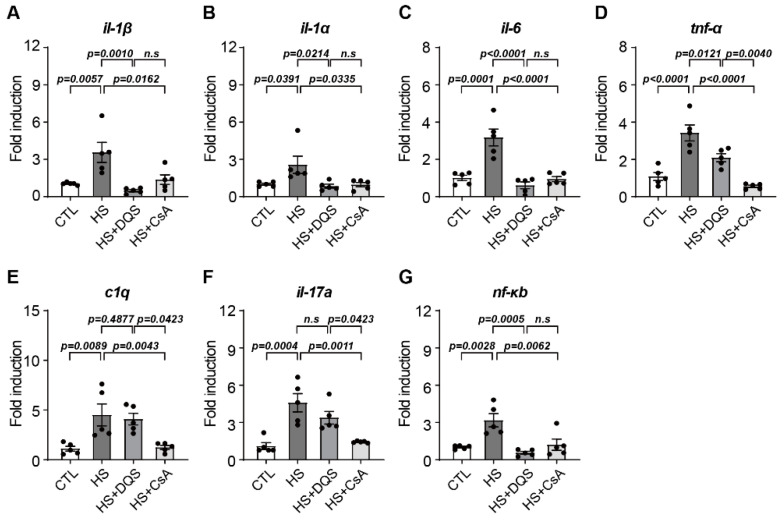
Changes in inflammatory cytokines under hyperosmotic conditions in HCECs treated with DQS and CsA. (**A**–**G**) The effect of DQS and CsA on inflammatory cytokine (*il-1β*, *il-1α*, *il-6*, *Tnf-α*, *c1q*, *il-17a*, and *nf-kb*) changes was investigated. HCECs were treated with CTL, CTL with 500 mOsm/L of NaCl (HS), HS with 100 µM of DQS (HS + DQS), and HS with 1 µM of CsA (HS + CsA) for 8 h. Quantitative PCR was performed to detect inflammatory cytokine mRNA expression. *gapdh* expression was used as a loading control (*n* = 5). Bars represent the mean ± S.E.M. *n.s*: nonsignificant.

**Figure 4 cells-13-01251-f004:**
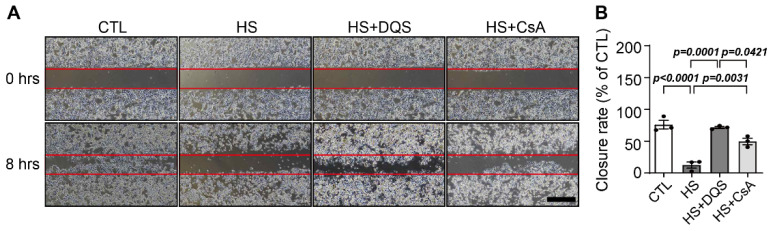
Effect of DQS and CsA in wound healing scratch assays. (**A**) HCECs were subjected to a scratch assay (scratch area; red box) and then treated with CTL, CTL with 500 mOsm/L of NaCl (HS), HS with 100 µM of DQS (HS + DQS), and HS with 1 µM of CsA (HS + CsA) for 8 h. Representative phase contrast images were acquired (scale bar = 50 μm). (**B**) The effect of DQS and CsA on the wound closure rate in the scratch assay was measured. The closure rate is shown as a percentage of that of the control cells. Bars represent the mean ± S.E.M. (*n* = 3).

**Figure 5 cells-13-01251-f005:**
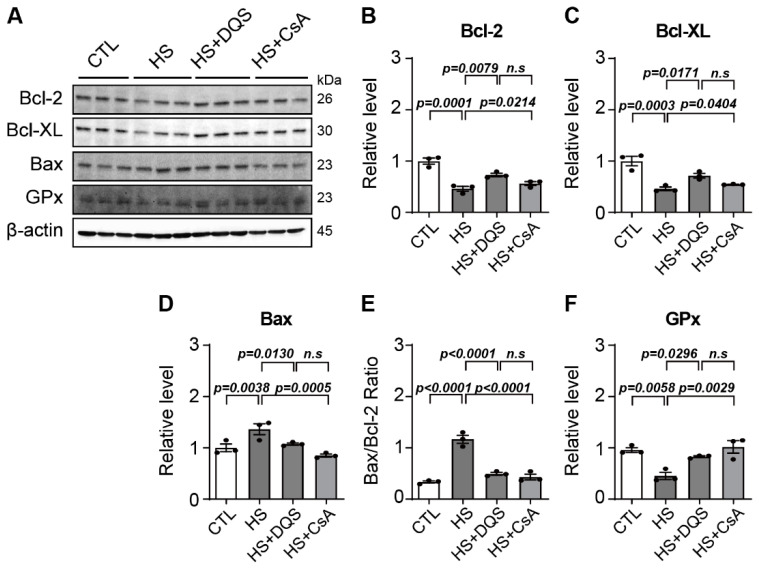
Changes in total cellular proteins under hyperosmotic conditions in HCECs with DQS and CsA. (**A**) Total cellular proteins were extracted from HCECs in a normal medium (CTL), CTL with 500 mOsm/L of NaCl (HS), DQS (HS + DQS; 100 μM), and CsA (HS + CsA; 1 μM) for 8 h. Protein levels of Bcl-2, Bcl-xL, Bax, and GPx were determined by Western blot analysis. Beta-actin was used as a control for protein loading. (**B**–**D**,**F**). Quantification of the expression levels of Bcl-2, Bcl-xl, Bax, and GPx in HCECs in the CTL, HS, HS+DQS, and HS+CsA groups (*n* = 3). (**E**) Bax and Bcl-2 protein concentrations were calculated by averaging the results of three independent experiments (*n* = 3). Bars represent the mean ± S.E.M. *n.s*: non-significant.

**Figure 6 cells-13-01251-f006:**
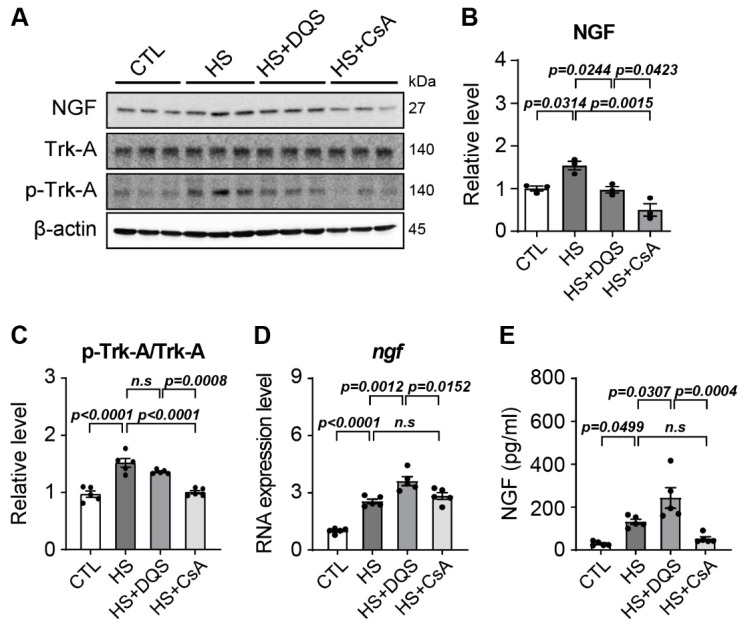
Changes in NGF protein expression in response to hyperosmotic conditions in HCECs treated with DQS and CsA. (**A**) Western blots evaluating the expression of NGF in HCECs exposed to hyperosmotic stress (HS) with DQS (HS + DQS; 100 μM) and CsA (HS + CsA; 1 μM). Beta-actin was used as a loading control. (**B**) Quantification of NGF expression levels in HCECs in the CTL, HS, HS + DQS, and HS+CsA groups (*n* = 3). (**C**) Quantification of the protein expression levels of p-Trk-A and t-Trk-A was calculated by averaging the results of three independent experiments (*n* = 5). (**D**) Quantitative PCR was performed to detect NGF expression. GAPDH expression was used as a loading control (*n* = 5). (**E**) NGF production was measured by ELISAs in a normal medium (CTL) and a medium with an osmolarity of 500 mOsm/L NaCl (HS), HS with DQS (HS + DQS; 100 μM) or CsA (HS + CsA; 1 μM) for 8 h (*n* = 5). Bars represent the mean ± S.E.M. *n.s*: nonsignificant.

**Figure 7 cells-13-01251-f007:**
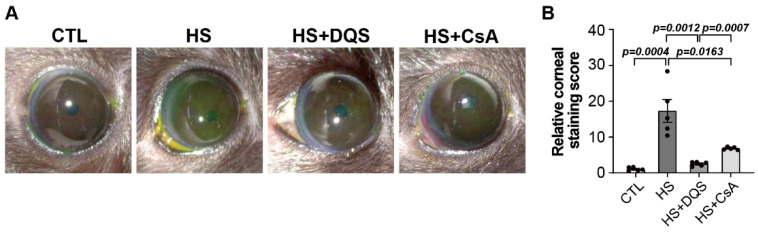
Corneal fluorescein staining after topical application of DQS or CsA. The experimental mice were subjected to hyperosmotic medium (500 mOsm/L; HS) or HS with 3% DQS or 0.05% CsA treatment for 10 days. (**A**) Photographs of corneal fluorescein staining in the four groups. (**B**) Clinical evaluation of corneal fluorescein staining. Quantification of corneal staining was calculated by averaging the results of five independent experiments. Bars represent the mean ± S.E.M.

**Figure 8 cells-13-01251-f008:**
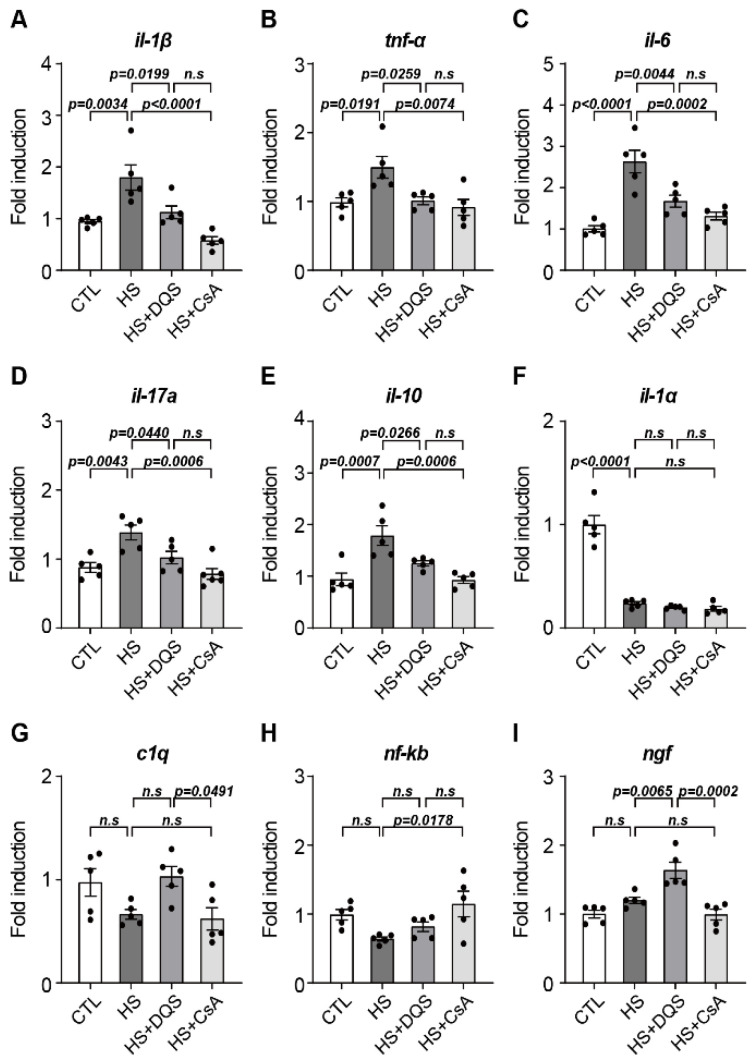
Changes in inflammatory cytokines and *ngf* under hyperosmotic conditions in corneas with DQS and CsA. (**A**–**I**) The effect of DQS and CsA on inflammatory cytokines (*il-1β*, *Tnf-α*, *il-6*, *il-17a*, *il-10*, *il-1α*, *c1q*, *nf-kb*) and *ngf* expression was investigated. Mouse corneas were treated with CTL, CTL with 500 mOsm/L NaCl (HS), HS with 3% DQS (HS + DQS), and HS with 0.05% CsA (HS + CsA) for 10 days. Quantitative PCR was performed to detect inflammatory cytokine (**A**–**H**) and *ngf* (**I**) mRNA expression. *gapdh* expression was used as a loading control (*n* = 5). Bars represent the mean ± S.E.M. *n.s*: nonsignificant.

**Figure 9 cells-13-01251-f009:**
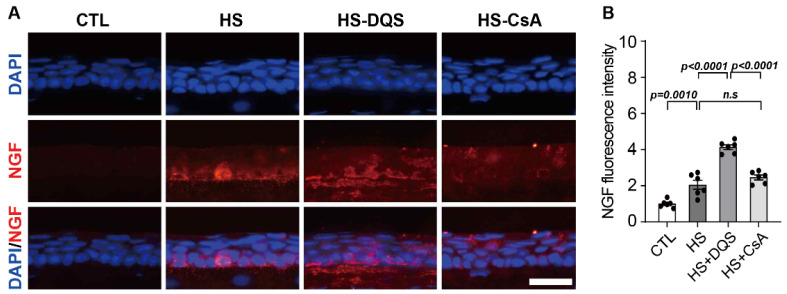
Effect of topical DQS and CsA on NGF expression in DED mice. The experimental mice were subjected to hyperosmotic medium (500 mOsm/L; HS) and HS with 3% DQS or 0.05% CsA treatment for 10 days. (**A**) Fluorescence immunocytochemistry for NGF and 4′,6-diamidino-2-phenylindole (DAPI) in the corneal epithelium. Scale bar length, 25 μM. (**B**) Quantification of NGF fluorescence intensity was calculated by averaging the results of five independent experiments (*n* = 6). Bars represent the mean ± S.E.M. *n.s*: nonsignificant.

**Table 1 cells-13-01251-t001:** Primer sequences used for mRNA expression analysis with gene name and sequence.

	Gene	Primer	Sequences (5′-3′)
1	*il-1β*	Forward	GCTCGCCAGTGAAATGATGG
		Reverse	GTCCTGGAAGGAGCACTTCAT
2	*il-1α*	Forward	GCACCTTACACCTACCAGAGT
		Reverse	AAACTTCTGCCTGACGAGCTT
3	*il-6*	Forward	TAGTCCTTCCTACCCCAATTTCC
		Reverse	TTGGTCCTTAGCCACTCCTTC
4	*Tnf-α*	Forward	CCCTCACACTCAGATCATCTTCT
		Reverse	GCTACGACGTGGGCTACAG
5	*il-17a*	Forward	CCCGGACTGTGATGGTCAAC
		Reverse	CGGTGGAGATTCCAAGGTGA
6	*il-10*	Forward	CTCCGAGATGCCTTCAGCAG
		Reverse	CTCAGACAAGGCTTGGCAAC
7	*c1q*	Forward	TCTGCACTGTACCCGGCTA
		Reverse	CCCTGGTAAATGTGACCCTTTT
8	*nf-kb*	Forward	TCCGTTATGTATGTGAAGGCCC
		Reverse	AACCTTTGCTGGTCCCACAT
9	*ngf*	Forward	ACAGGACTCACAGGAGCAAG
		Reverse	TCTTATCCCCAACCCACACG
10	*il-1β* (*mouse*)	Forward	GTTCCCATTAGACAACTGC
		Reverse	GATTCTTTCCTTTGAGGC
11	*il-1α* (*mouse*)	Forward	TCTGCCATTGACCATCTC
		Reverse	ATCTTCCCGTTGCTTGAC
12	*il-6* (*mouse*)	Forward	CTTGGGACTGATGCTGGTGACA
		Reverse	GCCTCCGACTTGTGAAGTGGTA
13	*Tnf-α* (*mouse*)	Forward	CCCTCACACTCAGATCATCTTCT
		Reverse	GCTACGACGTGGGCTACAG
14	*il-17a* (*mouse*)	Forward	CGCAATGAAGACCCTGATAG
		Reverse	TCCCTCCGCATTGACACA
15	*il-10* (*mouse*)	Forward	GCTCTTACTGACTGGCATGAG
		Reverse	CGCAGCTCTAGGAGCATGTG
16	*c1q* (*mouse*)	Forward	AAAGGCAATCCAGGCAATATCA
		Reverse	TGGTTCTGGTATGGACTCTCC
17	*nf-kb* (*mouse*)	Forward	TGCCAAAGAAGGACACGACA
		Reverse	AGGCTATTGCTCATCACAG
18	*ngf* (*mouse*)	Forward	TTGCCAAGGACGCAGCTTTC
		Reverse	TTCTGCCTGTACGCCGATC

## Data Availability

All datasets generated and analyzed during this study are included in this published article and its Appendix A. Additional data are available from the corresponding author upon reasonable request.

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
