# Peer review of "Diquafosol Improves Corneal Wound Healing by Inducing NGF Expression in an Experimental Dry Eye Model"

_cells, 2024, doi:10.3390/cells13151251_

Round 1
Reviewer 1 Report
Comments and Suggestions for Authors
Reviewer’s Comments
The manuscript titled ‘Diquafosol improves corneal wound healing by inducing NGF expression in an experimental dry eye model’ compared the therapeutic effects and mechanisms between two drugs, cyclosporine A (CsA) and diquafosol tetrasodium (DQS) for dry eye mitigation. The main results include cell viability, wound healing, and cornea damage as well as the suppression of inflammatory factors, including IL-1β, IL-1α, IL-6, TNF-α, C1q, and IL-17a. Differential effects were observed, particularly in the effect of Diquafosol on the increase of NGF expression and translocation into the extracellular space. The authors concluded that eye drops should be based on the therapeutic goals, either for cornea repair or inflammation inhibition, depending on the disease progression. The major contribution of the manuscript is the differential effects between the two drugs, particularly on the NGF expression.
While the manuscript adds novel insights into the therapeutic effects and mechanisms of the two drugs, some presentations are ambiguous and demand revision/re-wording.
1. In the Abstract: This study aims to compare the therapeutic mechanisms of cyclo sporine A (CsA) and diquafosol tetrasodium (DQS). The ‘among dry eye medications’ should be deleted.
2. In the Abstract: ‘Thus, depending on the type of lesion, eye drops should be selected according to the therapeutic target, focusing on repairing cellular damage when cellular repair is needed or reducing inflammation when inflammation is high and cellular damage is severe.’ The present data do not support this conclusion. In the results, both CsA and DQS were beneficial in wound healing and reducing inflammation, how can they be differentially used based on therapeutic target? In clinical dry eye, cornea epithelial damage is concomitant with inflammation almost all the time. Would it be practical to separate the therapeutic target?
3. Line 70: The study aims included how CsA and AQS may influence the ‘development of corneal opacity.’ However, no data was shown regarding the cornea opacity. The cornea staining reflects the ocular surface damage, not cornea opacity.
4. Line 225: The cell line data cannot be regarded as equivalent to in vivo dry eye status. The ‘experimental dry eye and cell damage’ should be revised as ‘ cell damage’.
5. Line 316: The figure legends do not match with the data.
6. Line 327: ‘Figure 7. Topical application of DQS and CsA was confirmed by corneal fluorescein staining’? How can the topical application be confirmed by corneal fluorescein staining? The title is suggested to be recast as ‘Figure 7. Corneal fluorescein staining after topic application of DQS or CsA’.
7. Line 316, Figure 6: A discrepancy can be found in cellular NGF (shown by Western blot) and NGF in the medium (shown by ELISA). The HS group displayed the highest cellular NGF content by Western Blot analysis, while less NGF was detected by ELISA in the medium. NGF RNA expression levels did not agree with the Western Blot data. This should be explained and brought in the Discussion.
8. Line 363: ‘the expressionlevels were’ should be revised as ‘ the expression levels were…’.
9. Line 382: ‘Laibovitz et al.’ may better be revised as ‘Laibovitz et al. [5]’
10. Line 484: Informed Consent Statement: Informed consent was obtained from all subjects involved in the study. There is no way to obtain informed consent from a cell line or mice.
11. Line 489: The authors should specify acknowledgment or just delete it.
Comments on the Quality of English LanguageSome presentations are ambiguous and demand revision/re-wording.
Author Response
Response to Reviewer 1 Comments
The manuscript titled ‘Diquafosol improves corneal wound healing by inducing NGF expression in an experimental dry eye model’ compared the therapeutic effects and mechanisms between two drugs, cyclosporine A (CsA) and diquafosol tetrasodium (DQS) for dry eye mitigation. The main results include cell viability, wound healing, and cornea damage as well as the suppression of inflammatory factors, including IL-1β, IL-1α, IL-6, TNF-α, C1q, and IL-17a. Differential effects were observed, particularly in the effect of Diquafosol on the increase of NGF expression and translocation into the extracellular space. The authors concluded that eye drops should be based on the therapeutic goals, either for cornea repair or inflammation inhibition, depending on the disease progression. The major contribution of the manuscript is the differential effects between the two drugs, particularly on the NGF expression.
While the manuscript adds novel insights into the therapeutic effects and mechanisms of the two drugs, some presentations are ambiguous and demand revision/re-wording.
Point 1 : In the Abstract: This study aims to compare the therapeutic mechanisms of cyclosporine A (CsA) and diquafosol tetrasodium (DQS). The ‘among dry eye medications’ should be deleted.
Response 1 : We have made the revisions as per your comments.
Point 2 : In the Abstract: ‘Thus, depending on the type of lesion, eye drops should be selected according to the therapeutic target, focusing on repairing cellular damage when cellular repair is needed or reducing inflammation when inflammation is high and cellular damage is severe.’ The present data do not support this conclusion. In the results, both CsA and DQS were beneficial in wound healing and reducing inflammation, how can they be differentially used based on therapeutic target? In clinical dry eye, cornea epithelial damage is concomitant with inflammation almost all the time. Would it be practical to separate the therapeutic target?
Response 2 : Thank you for the detailed reviewer's comments. As you mentioned, when treating patients with dry eye syndrome, inflammation and corneal epithelial damage are often concomitant. In fact, there are reports that conjunctival and corneal epithelial erosion can be indicators of the severity of inflammation. In countries like the United States and Japan, where only one drug such as DQS or CsA can be used for the treatment of dry eye syndrome, there is no choice but to treat dry eye syndrome with one of the two medications that can be prescribed. However, in countries like Korea, where both drugs are available, it becomes a matter of consideration which drug to choose as the initial treatment. In such cases, our study suggests the theoretical background for selecting CsA when conjunctival and corneal inflammation predominates, such as in Sjogren’s syndrome or other rheumatic diseases. On the other hand, when the MMP-9 test and other indicators are not particularly high, but ocular surface erosion is severe, DQS might be considered as a treatment option.
Point 3 : Line 70: The study aims included how CsA and AQS may influence the ‘development of corneal opacity.’ However, no data was shown regarding the cornea opacity. The cornea staining reflects the ocular surface damage, not cornea opacity.
Response 3 : As you pointed out, corneal opacity was not identified in this study; only corneal surface damage could be identified through staining. We changed corneal opacity to corneal surface damage as you recommended.
Point 4 : Line 225: The cell line data cannot be regarded as equivalent to in vivo dry eye status. The ‘experimental dry eye and cell damage’ should be revised as ‘ cell damage’.
Response 4 : We have made the revisions as you mentioned.
Point 5 : Line 316: The figure legends do not match with the data.
Response 5 : We have revised it to ensure the figure legend and data match accurately.
Point 6 : Line 327: ‘Figure 7. Topical application of DQS and CsA was confirmed by corneal fluorescein staining’? How can the topical application be confirmed by corneal fluorescein staining? The title is suggested to be recast as ‘Figure 7. Corneal fluorescein staining after topic application of DQS or CsA’.
Response 6 : Your point is correct. We have made the changes as per your suggestion.
Point 7 : Line 316, Figure 6: A discrepancy can be found in cellular NGF (shown by Western blot) and NGF in the medium (shown by ELISA). The HS group displayed the highest cellular NGF content by Western Blot analysis, while less NGF was detected by ELISA in the medium. NGF RNA expression levels did not agree with the Western Blot data. This should be explained and brought in the Discussion.
Response 7 : Thank you for your very important point. The authors were also aware of the issue raised by the reviewer and have carefully considered its interpretation. The authors confirmed that intracellular NGF RNA expression increases due to HS stress, but translocation out of the cell is less in the group treated with DQS. We believe that DQS influences the translocation of NGF. Further research on the detailed mechanism will be conducted in the future. The authors have described this content in the discussion section as follows.
Point 8 : Line 363: ‘the expressionlevels were’ should be revised as ‘ the expression levels were…’.
Response 8 :We have made the revisions according to your suggestions.
Point 9 : Line 382: ‘Laibovitz et al.’ may better be revised as ‘Laibovitz et al. [5]’
Response 9 : Thank you for your comments. We have made the revisions according to your suggestions.
Point 10 :. Line 484: Informed Consent Statement: Informed consent was obtained from all subjects involved in the study. There is no way to obtain informed consent from a cell line or mice.
Response 10 : We have deleted the above phrase according to your suggestion.
Point 11 : Line 489: The authors should specify acknowledgment or just delete it.
Response 11 : We have deleted the above acknowledgements according to your suggestion.

Reviewer 2 Report
Comments and Suggestions for Authors
This experimental study examines the mechanisms by which cyclosporine A and diquafosol tetrasodium influence corneal wound healing. The authors compared their effects on inflammatory cytokine expression, apoptosis markers and nerve growth factor signalling. They have used an in vivo experiment on the dry eye mouse model and an in vitro model to demonstrate the cell viability and wound healing mechanism. The introduction and methods sections are well written and it adequately covers the background and the experiment. The authors noted the distinct effects of cyclosporine A and diquafosol tetrasodium on the specific cytokines. They highlighted the role of nerve growth factor in the corneal wound healing process. Based on their findings, they concluded that the clinical applications of cyclosporine A and diquafosol tetrasodium should be differentiated based on the underlying pathophysiology and phenotype of the dry eye disease. They have also suggested future research to explore combination therapies and personalised treatment approaches to optimize treatment outcomes. It is interesting to find that the authors used the term “lesion” in the abstract, but this term is not used in the main text of the manuscript. For consistency, the authors should try to maintain the same terminologies in both the abstract and the main text of the manuscript.
Author Response
Response to Reviewer 2 Comments
This experimental study examines the mechanisms by which cyclosporine A and diquafosol tetrasodium influence corneal wound healing. The authors compared their effects on inflammatory cytokine expression, apoptosis markers and nerve growth factor signalling. They have used an in vivo experiment on the dry eye mouse model and an in vitro model to demonstrate the cell viability and wound healing mechanism. The introduction and methods sections are well written and it adequately covers the background and the experiment. The authors noted the distinct effects of cyclosporine A and diquafosol tetrasodium on the specific cytokines. They highlighted the role of nerve growth factor in the corneal wound healing process. Based on their findings, they concluded that the clinical applications of cyclosporine A and diquafosol tetrasodium should be differentiated based on the underlying pathophysiology and phenotype of the dry eye disease. They have also suggested future research to explore combination therapies and personalised treatment approaches to optimize treatment outcomes. It is interesting to find that the authors used the term “lesion” in the abstract, but this term is not used in the main text of the manuscript. For consistency, the authors should try to maintain the same terminologies in both the abstract and the main text of the manuscript.
Response : Thank you for your positive reviewer comments. As you pointed out, 'clinical phenotypes' is a more appropriate term than 'lesion', and we have revised the manuscript accordingly.

Reviewer 3 Report
Comments and Suggestions for Authors
The manuscript by Song et al. describes a study on the beneficial effects of DQS in enhancing the healing of corneal wounds through the induction of NGF expression in a dry eye experimental model. The authors provide evidence that both CsA and DQS have the ability to regulate several cellular processes, including cell viability, wound healing, apoptosis, and damage to the cornea caused by hyperosmolarity. Furthermore, the study reveals that DQS can increase NGF expression and facilitate its translocation into the extracellular space.
The authors are commended for an interesting study. The manuscript is well-organized and conclusive. Specific comments/suggestions to further improve the manuscript before acceptance are as follows:
· In general
1. The authors should revise the manuscript to ensure that all references are correctly and appropriately cited.
2. The writing of the manuscript should be improved.
· Introduction
1. It would benefit a broader audience if the authors could incorporate a basic introduction outlining the symptoms associated with dry eye.
2. The authors are encouraged to provide a comprehensive introduction to the existing treatments for dry eye, along with their potential drawbacks.
· Results
1. The authors should revisit and clarify the presentation of their results. Some of the findings were not articulated clearly in the current version.
2. Figure 1A. The clarity of the images in the manuscript could be improved. Please add images with a higher magnification or provide zoomed-in versions for better visibility.
3. The authors have observed differential responses in certain inflammatory cytokines to CsA and DQS. Would it be possible to consider a combined treatment approach using both CsA and DQS?
· Discussion
1. In Figure 2, the authors demonstrated that high concentrations of CsA and DQS led to a reduction in cell viability. Could you please discuss this observation?
2. Could the authors discuss any potential limitations and future directions that have been identified in this study?
Comments on the Quality of English LanguageThe writing of the manuscript should be improved.
Author Response
Response to Reviewer 3 Comments
The manuscript by Song et al. describes a study on the beneficial effects of DQS in enhancing the healing of corneal wounds through the induction of NGF expression in a dry eye experimental model. The authors provide evidence that both CsA and DQS have the ability to regulate several cellular processes, including cell viability, wound healing, apoptosis, and damage to the cornea caused by hyperosmolarity. Furthermore, the study reveals that DQS can increase NGF expression and facilitate its translocation into the extracellular space.
The authors are commended for an interesting study. The manuscript is well-organized and conclusive. Specific comments/suggestions to further improve the manuscript before acceptance are as follows:
Point 1 : The authors should revise the manuscript to ensure that all references are correctly and appropriately cited.
Response 1 : We have reviewed all the references to ensure accurate citation and made necessary corrections
Point 2 : The writing of the manuscript should be improved.
Response 2 : We have carefully reviewed the manuscript and revised some inappropriate words or expressions.
- Introduction
Point 3 : It would benefit a broader audience if the authors could incorporate a basic introduction outlining the symptoms associated with dry eye.
Response 3 : As the reviewer pointed out, we have added a description of the general symptoms of dry eye disease in the introduction section as follows.
Point 4 : The authors are encouraged to provide a comprehensive introduction to the existing treatments for dry eye, along with their potential drawbacks.
Response 4 : As the reviewer suggested, we have added a description of the various types of medications used for dry eye disease and their limitations in the introduction section as follows.
- Results
Point 5 : The authors should revisit and clarify the presentation of their results. Some of the findings were not articulated clearly in the current version.
Response 5 : As the reviewer pointed out, we have made revisions to present the results accurately.
Point 6 : Figure 1A. The clarity of the images in the manuscript could be improved. Please add images with a higher magnification or provide zoomed-in versions for better visibility.
Response 6 : As the reviewer pointed out, we have revised the image in Fig 1A to have higher resolution and magnification
Point 7 : The authors have observed differential responses in certain inflammatory cytokines to CsA and DQS. Would it be possible to consider a combined treatment approach using both CsA and DQS?
Response 7 : As you know, when treating patients with dry eye syndrome, inflammation and corneal epithelial damage are often concomitant. In fact, there are reports that conjunctival and corneal epithelial erosion can be indicators of the severity of inflammation. In countries like the United States and Japan, where only one drug such as DQS or CsA can be used for the treatment of dry eye syndrome, there is no choice but to treat dry eye syndrome with one of the two medications that can be prescribed. However, in countries like Korea, where both drugs are available, it becomes a matter of consideration which drug to choose as the initial treatment. In such cases, our study suggests the theoretical background for selecting CsA when conjunctival and corneal inflammation predominates, such as in Sjogren’s syndrome or other rheumatic diseases. On the other hand, when the MMP-9 test and other indicators are not particularly high, but ocular surface erosion is severe, DQS might be considered as a treatment option.As the reviewer mentioned, the authors also believe that using both medications together would be more effective for certain dry eye disease patients.
- Discussion
Point 8 : In Figure 2, the authors demonstrated that high concentrations of CsA and DQS led to a reduction in cell viability. Could you please discuss this observation?
Response 8 :As previously published papers have stated, both CsA and DQS decrease the cell viability of HCECs at higher concentrations. We conducted our experiments using doses that, according to previous studies, do not significantly affect cell viability.
Point 9 : Could the authors discuss any potential limitations and future directions that have been identified in this study?
Response 9 : As the reviewer pointed out, we have additionally described the limitations of our study and the directions for future research as follows.;
“However, our study has the following limitations. First, we used a corneal epithelial cell line. Second, we used a hyperosmolar solution as an in vivo dry eye model. Third, we did not investigate the mechanism by which DQS facilitates the secretion of increased NGF expression from HCECs. Future research will utilize primary cultured cells and well-known mouse models such as desiccating stress or scopolamine injection to study the mechanisms involved.”

Round 2
Reviewer 1 Report
Comments and Suggestions for Authors
The authors have made corrections and responded to the points raised by the reviewer.